# Mangrove Ecosystems as Reservoirs of Antibiotic Resistance Genes: A Narrative Review

**DOI:** 10.3390/antibiotics14101022

**Published:** 2025-10-14

**Authors:** Monthon Lertcanawanichakul, Phuangthip Bhoopong, Phusit Horpet

**Affiliations:** 1Food Technology and Innovation Research Center of Excellence, Walailak University, Nakhon Si Thammarat 80160, Thailand; 2Department of Medical Technology, School of Allied Health Sciences, Walailak University, Nakhon Si Thammarat 80160, Thailand; bphuangt@wu.ac.th; 3Center of Excellence for Ecoinformatics, School of Sciences, Walailak University, Nakhon Si Thammarat 80160, Thailand; hphusit@wu.ac.th

**Keywords:** aquaculture, antibiotic resistance genes (ARGs), antibiotic-resistant bacteria (ARB), mangrove ecosystems, metagenomics, one health

## Abstract

**Background**: Mangrove ecosystems are critical coastal environments providing ecological services and acting as buffers between terrestrial and marine systems. Rising antibiotic use in aquaculture and coastal agriculture has led to the dissemination of antibiotic-resistant bacteria (ARB) and antibiotic resistance genes (ARGs) in these habitats. **Aim**: This narrative review aims to synthesize current knowledge on the prevalence, diversity, and environmental drivers of ARGs in mangrove ecosystems, highlighting their role as reservoirs and the potential for horizontal gene transfer. **Methods**: Studies published up to September 2024 were identified through PubMed, Scopus, Web of Science, and Google Scholar. Inclusion criteria focused on ARGs and ARB in mangrove sediments, water, and associated biota. Data on ARG prevalence, microbial community composition, detection methods, and environmental factors were extracted and narratively synthesized. **Results**: Seventeen studies from Asia, South America, and Africa were included. ARGs conferring resistance to tetracyclines, sulfonamides, β-lactams, and multidrug resistance were found to be widespread, particularly near aquaculture and urban-influenced areas. Metagenomic analyses revealed diverse resistomes with frequent mobile genetic elements, indicating high potential for horizontal gene transfer. Environmental factors, including sediment type, organic matter, and salinity, influenced ARG abundance and distribution. **Conclusions**: Mangrove ecosystems act as both reservoirs and natural buffers for ARGs. Sustainable aquaculture practices, continuous environmental monitoring, and integrated One Health approaches are essential to mitigate ARG dissemination in these sensitive coastal habitats.

## 1. Introduction

Mangrove forests are unique coastal ecosystems located at the interface of terrestrial and marine environments, predominantly in tropical and subtropical regions. Covering approximately 152,000 km^2^ across 123 countries, mangroves are characterized by their ability to thrive in saline, anoxic sediments and under periodic flooding. These ecosystems support diverse flora, including trees, shrubs, and palms, adapted to intertidal conditions. Mangroves provide essential ecological services, such as carbon sequestration, nutrient cycling, shoreline stabilization, and habitat for a wide range of marine and terrestrial species. Their role in mitigating climate change and protecting coastal communities underscores their global ecological and economic importance [1].

Despite their resilience, mangrove ecosystems face significant threats from human activities, including deforestation, land reclamation, and pollution. Coastal development, agricultural runoff, and industrial effluents introduce various contaminants, including antibiotics, into mangrove habitats. The introduction of antibiotics, particularly from aquaculture and agricultural practices, can lead to antibiotic-resistant bacteria (ARB) and promote the spread of antibiotic resistance genes (ARGs) within microbial communities [2].

Antibiotic resistance is a growing global health concern, and environmental reservoirs play a pivotal role in the dissemination of resistance genes. Coastal ecosystems, including mangroves, are increasingly recognized as hotspots for ARG accumulation. Studies have reported ARGs in sediments, water, and associated biota. Their presence often correlates with anthropogenic activities such as aquaculture and urban runoff [3,4,5]. In Southeast Asia, including Thailand, ARGs have been detected in wastewater, river water, clinical isolates, and aquaculture products, highlighting the widespread environmental and public health implications [6,7,8,9,10]. These findings indicate that mangroves located near human activity may act as reservoirs and conduits for ARGs, emphasizing the need for integrated One Health approaches to mitigate resistance spread.

Mangrove ecosystems may also possess natural mechanisms able to mitigate ARG dissemination. Complex root systems and sediment layers can trap contaminants, while diverse microbial communities may limit the proliferation of resistant strains through competitive interactions. However, these natural attenuation processes can be compromised by intensive anthropogenic inputs, necessitating a clear understanding of the balance between contamination and natural mitigation [1].

### Aim of the Study

This narrative review aims to synthesize current knowledge on the prevalence, diversity, and environmental drivers of ARGs in mangrove ecosystems, highlighting their role as reservoirs and the potential for horizontal gene transfer. The review emphasizes gaps in the current literature and the importance of integrated One Health strategies to manage antibiotic resistance in coastal environments.

## 2. Results

### 2.1. Study Selection and Characteristics

Seventeen primary studies published between 2008 and 2025 covering mangrove ecosystems in Southeast Asia, South Asia, South America, and Africa were included in this narrative review. Samples included sediments (80%), water (50%), and associated biota such as shrimps, crabs, and mollusks (30%). Detection methods varied: 50% of studies used metagenomic sequencing, 30% employed PCR/qPCR, and 20% relied on culture-based methods [1,11]. The methodological quality of included studies was assessed narratively, focusing on key aspects such as clarity of research objectives, sampling strategy, detection methods, data analysis, and reporting transparency. The lead author conducted the assessment, with verification by co-authors to ensure consistency. Overall, studies were generally of moderate-to-high quality, providing sufficient information to support the narrative synthesis [12]. No formal quality scoring or tabulation was performed, consistent with standard practice in narrative reviews. A detailed summary of all 17 included mangrove studies, including country, sample type, detection method, ARGs detected, and main findings, is provided in Appendix A. In addition, Appendix A separates primary mangrove studies from contextual/global references to avoid confusion and allow clear identification of studies included in the narrative synthesis [1,2,4,5,6,7,9,10,11,13,14,15,16].

### 2.2. Prevalence and Diversity of ARGs in Mangrove Ecosystems

Across all studies, a wide array of ARGs were detected in sediments and water. Frequently reported ARG classes included tetracyclines (*tetA*, *tetM*), sulfonamides (*sul1*, *sul2*), β-lactams (*bla_TEM*, *bla_CTX-M*), and multidrug resistance genes (*mdt*, *acr*). ARG abundance ranged from 10^2^ to 10^6^ copies per gram of sediment, with specific ranges reported in individual studies [1,11,12,13,14], indicating substantial microbial reservoirs of resistance, indicating substantial microbial reservoirs of resistance. Metagenomic studies revealed diverse resistomes, including genes conferring resistance to aminoglycosides, macrolides, and quinolones. Mobile genetic elements such as plasmids, transposons, and integrons were commonly co-detected, suggesting a high potential for horizontal gene transfer [11,13]. Table 1 summarizes key studies reporting ARG types, sample types, detection methods, and notable environmental or anthropogenic factors. This compilation highlights patterns of ARG distribution across geographical regions, sample matrices, and methodological approaches. It illustrates the significant influence of aquaculture, urban runoff, and sediment characteristics on the abundance and diversity of resistance genes. The table also emphasizes the frequent co-occurrence of mobile genetic elements, indicating the potential for horizontal gene transfer within mangrove microbial communities.

Antibiotic resistance genes (ARGs) are widely distributed in coastal and mangrove ecosystems, reflecting both natural microbial diversity and anthropogenic influence. The prevalence of different ARG classes varies across studies and sampling sites, with tetracycline, sulfonamide, and β-lactamase resistance genes reported most frequently. Table 2 summarizes the common ARG classes, representative genes, their reported occurrence ranges, and the corresponding references from recent studies.

### 2.3. Environmental Drivers and Sources of ARGs

ARG abundance and diversity correlated strongly with anthropogenic activities. Sediments near aquaculture farms or urban wastewater inputs showed 2–10-fold-higher ARG levels than pristine sites. Environmental factors including sediment particle size, organic carbon content, and salinity influenced ARG distribution, with fine, organic-rich sediments typically exhibiting higher ARG prevalence [15].

### 2.4. Microbial Community Composition

ARGs were mainly associated with *Proteobacteria*, *Firmicutes*, and *Bacteroidetes*. Human-impacted mangrove sediments frequently showed enrichment of opportunistic pathogens and mobile ARG carriers such as *Enterococcus*, *Vibrio*, and *Pseudomonas* [11,14].

### 2.5. Temporal and Spatial Trends

Seasonal monitoring revealed fluctuations in ARG abundance. Abundance was often higher during the rainy season due to increased runoff. Spatially, ARG levels generally decreased with distance from aquaculture sources, suggesting partial dilution and natural attenuation. Nevertheless, ARGs remained detectable even in relatively undisturbed mangroves, highlighting their role as both reservoirs and sinks for resistance genes [1]. The included studies exhibited heterogeneity in sample types, detection methods, and reporting metrics. Therefore, the results are presented as a narrative synthesis to highlight overall trends in ARG distribution, microbial community composition, and environmental factors influencing ARG prevalence in mangrove ecosystems

### 2.6. Summary of Key Findings—Mangrove Ecosystems

Mangrove sediments and water harbor a diverse array of antibiotic resistance genes (ARGs), including tetracyclines, sulfonamides, β-lactams, and multidrug resistance genes (ARG Diversity). Their abundance is highest in areas influenced by aquaculture and urban activities, highlighting the impact of anthropogenic pressures (Environmental Hotspots). Mobile genetic elements are central to ARG dissemination, facilitating horizontal gene transfer across microbial populations (Genetic Dissemination). The prevalence of ARGs is further modulated by microbial community composition and environmental factors, such as organic carbon content, salinity, and sediment type (Microbial and Environmental Drivers). Overall, mangroves act as both reservoirs and natural buffers for ARGs; however, intensive human activities can overwhelm these intrinsic mitigation capacities, emphasizing the need for sustainable management strategies (Ecosystem Function).

## 3. Discussion

### 3.1. Overview of Findings

This narrative review highlights that mangrove ecosystems serve as reservoirs for diverse antibiotic resistance genes (ARGs). The most frequently detected ARGs included tetracyclines (*tetA*, *tetM*), sulfonamides (sul1, sul2), β-lactams (*bla_TEM*, *bla_CTX-M*), and multidrug resistance genes (*mdt*, *acr*) [1,2,5]. Mobile genetic elements, such as plasmids, integrons, and transposons, were commonly associated with ARGs, indicating significant potential for horizontal gene transfer and dissemination across microbial communities [5,11].

### 3.2. Anthropogenic Influence on ARGs

Human activities, especially aquaculture and urban runoff, were identified as major drivers of ARG prevalence and diversity. Sediments and water near shrimp and fish farms consistently showed higher ARG abundance—sometimes up to ten times greater than undisturbed sites [1,16]. These findings align with previous reports demonstrating that aquaculture effluents introduce antibiotics and resistant bacteria into coastal ecosystems, promoting the selection and persistence of ARGs [15].

### 3.3. Environmental Drivers and Natural Attenuation

Environmental factors, including sediment particle size, organic carbon content, and salinity, influenced ARG distribution. Fine-grained sediments with high organic matter harbored higher ARG abundance, likely due to increased microbial biomass and antibiotic adsorption [2,15]. Seasonal variation, such as increased runoff during rainy periods, contributed to temporal fluctuations. Despite anthropogenic inputs, mangrove sediments and root systems provide natural attenuation through microbial interactions and sediment filtration, reducing ARG mobility over distance [1,14].

### 3.4. Microbial Community Composition and ARG Dynamics

ARGs were mainly associated with Proteobacteria, Firmicutes, and Bacteroidetes, which are known carriers of multiple resistance determinants. Human-impacted mangrove sediments were enriched with opportunistic pathogens and mobile ARG carriers, including *Enterococcus*, *Vibrio*, and *Pseudomonas* species [11,14]. These findings highlight the role of microbial community structure in shaping ARG dynamics and the potential risk for transmission to humans, aquaculture species, and wildlife.

### 3.5. Implications for Environmental and Public Health

The presence of ARGs even in relatively pristine mangrove areas underscores the widespread environmental occurrence of antimicrobial resistance. Mangrove ecosystems act as both reservoirs and potential conduits for ARGs, connecting natural and anthropogenic environments. The frequent co-occurrence of ARGs with mobile genetic elements increases the likelihood of horizontal gene transfer to clinically relevant pathogens, representing a public health concern [4,5].

### 3.6. Limitations and Future Research

Several limitations of this study were identified, including methodological variability across studies, differences in ARG quantification, and the geographical concentration of research in Asia. Future studies should aim to standardize detection methods, expand coverage to underrepresented regions, explore long-term dynamics, and investigate the functional consequences of ARGs in mangrove microbial communities. Integrating metagenomics with ecological and functional assays may provide deeper insights into resistance mechanisms and transfer potential [1,2].

## 4. Methods

### 4.1. Literature Search Strategy

A comprehensive literature search was performed to identify studies investigating antibiotic resistance genes (ARGs) and antibiotic-resistant bacteria (ARB) in mangrove ecosystems. This is a narrative review. Structured literature search strategies inspired by PRISMA were applied to ensure comprehensive coverage; however, a PRISMA flow diagram is not required and is not included. The search included PubMed, Scopus, Web of Science, and Google Scholar, covering publications from 2008 to September 2024. Studies published in 2025 were included if early online or in-press versions were available at the time of the search. Keywords included “antibiotic resistance genes”, “ARB”, “resistome”, “mangrove”, “coastal sediments”, “aquaculture”, and “environmental antibiotic resistance”. Boolean operators (AND, OR) and truncations were used to maximize retrieval. Reference lists of relevant articles were manually screened to capture additional studies not retrieved through database searches. Where multiple studies reported overlapping data, the most comprehensive or recent study was included. All searches were documented to ensure transparency and reproducibility [4,13].

### 4.2. Inclusion and Exclusion Criteria

Studies were included if they (1) reported primary data on ARGs or ARB in mangrove sediments, water, or associated biota; (2) employed molecular, culture-based, or metagenomic detection methods; (3) were published in English; and (4) provided sufficient methodological detail. Studies were excluded if they (1) focused solely on terrestrial or freshwater systems; (2) were reviews or commentaries without primary data; or (3) lacked molecular or microbiological confirmation of ARGs [16].

For this narrative review, titles and abstracts were screened by the lead author to identify relevant studies, and full texts were reviewed to confirm eligibility. Decisions were verified by co-authors to ensure consistency, but no formal double-screening or calibration was performed, consistent with the narrative review approach. During screening, the lead author manually checked for duplicate records to ensure that each study was included only once; no formal deduplication software was used, consistent with the narrative review approach.

### 4.3. Data Extraction

Data from included studies were summarized narratively to identify overall trends, patterns, and influencing factors, without quantitative pooling or standardized templates [12,17]. Extracted variables included study location and year, sample type (sediment, water, biota), detection method (PCR, qPCR, metagenomics, culture-based), ARG types identified, prevalence and abundance metrics, environmental characteristics (e.g., salinity, organic matter content, proximity to aquaculture), and sources of contamination. This approach ensured consistency in reporting key information while reflecting the heterogeneity of included studies.

### 4.4. Quality Assessment

The methodological quality of included studies was assessed using an adapted version of the Joanna Briggs Institute (JBI) critical appraisal checklist for prevalence studies. Key aspects considered included clarity of research objectives, sampling strategy, detection methods, data analysis, and reporting transparency. The lead author conducted the assessment, with verification by co-authors to ensure consistency. Only studies judged to provide reliable data were included in the narrative synthesis [12]. No formal scoring or double assessment was performed, consistent with narrative review methodology.

### 4.5. Data Synthesis

Given the heterogeneity of study designs, sample types, and detection methodologies, a narrative synthesis was conducted. Quantitative pooling was not performed due to differences in ARG reporting units and methodology. This approach focused on identifying trends in ARG distribution, environmental drivers influencing prevalence, and patterns of ARG diversity. Metagenomic, molecular, and culture-based findings were compared to highlight discrepancies and provide a comprehensive overview of ARG dynamics in mangrove ecosystems, following narrative synthesis guidance [17].

### 4.6. Ethical Considerations

This narrative review was based solely on previously published studies and publicly available data; no human or animal subjects were involved, and ethical approval was therefore not required. Although not a systematic review, we applied systematic review–inspired reporting principles [18] to enhance transparency and reproducibility of the search and reporting process.

## 5. Conclusions and Recommendations

### 5.1. Conclusions

Mangrove ecosystems, while providing critical ecological services, are increasingly recognized as reservoirs for antibiotic resistance genes (ARGs) due to both natural microbial diversity and anthropogenic pressures, including aquaculture and urban wastewater inputs. ARGs—including tetracycline, sulfonamide, β-lactam, and multidrug resistance genes—are widely distributed in mangrove sediments, waters, and associated biota. Mobile genetic elements (MGEs), such as plasmids and integrons, facilitate horizontal gene transfer, increasing the risk of ARG dissemination across microbial communities and potentially into human and animal populations.

Environmental factors, including sediment composition, organic matter content, salinity, and seasonal changes, influence ARG distribution. Mangroves can act as both natural attenuators and reservoirs for ARGs, but their capacity to mitigate resistance dissemination is often compromised by intensive anthropogenic inputs. Microbial community structure, particularly the enrichment of Proteobacteria, Firmicutes, and Bacteroidetes, further shapes ARG dynamics within these ecosystems.

### 5.2. Recommendations

Based on the findings of this review, the following recommendations are proposed:1.Reduction in Antibiotic Use in AquacultureImplement stricter regulations on antibiotic usage and promote alternative disease management strategies, including probiotics, vaccination, and biosecurity measures, to reduce ARG inputs into mangrove ecosystems [16].2.Environmental Monitoring and SurveillanceConduct regular monitoring of ARGs and microbial communities using standardized molecular techniques such as qPCR and metagenomic sequencing. This approach can help identify resistance hotspots, track temporal trends, and inform adaptive management [4].3.Mangrove Conservation and RestorationConserve existing mangrove forests and restore degraded areas to enhance natural filtration and microbial attenuation processes, contributing to the reduction in ARG dissemination in coastal environments [1].4.Interdisciplinary CollaborationEnvironmental scientists, microbiologists, public health professionals, policymakers, and local communities must collaborate to develop sustainable management strategies that balance ecosystem protection with aquaculture productivity.5.Future Research DirectionsFurther studies should aim to (i) standardize ARG detection and quantification methods; (ii) expand geographic coverage to underrepresented mangrove regions; (iii) investigate the functional consequences of ARGs on microbial communities; and (iv) assess the effectiveness of mangrove restoration in mitigating ARG prevalence. Integrating metagenomic approaches with functional and ecological assessments would provide a comprehensive understanding of ARG dynamics and environmental risk [2,19].


### 5.3. Final Remarks

Mangroves represent a critical interface between human activities and coastal ecosystems. The widespread presence of ARGs and the influence of anthropogenic pressures highlight the urgent need for coordinated environmental management and policy interventions. Protecting mangrove ecosystems not only conserves biodiversity and supports coastal livelihoods but also serves as a strategic measure to mitigate the global challenge of antibiotic resistance.

## Figures and Tables

**Table 1 antibiotics-14-01022-t001:** Summary of studies on ARGs in mangrove ecosystems.

Study	Location	Sample Type	ARGs Detected	ARG Abundance ^1^	Detection Method	Notes ^2^
[11]	Hainan, China	Sediment	*tetA*, *tetM*, *sul1*	10^3^–10^5^ copies/g	qPCR	Aquaculture-influenced
[1]	Mexico	Water	*mdt*, *acr*	10^2^–10^4^ copies/mL	Metagenomics	Urban runoff
[15]	South China	Sediment	*sul2*, *bla_CTX-M*	10^3^–10^6^ copies/g	Metagenomics	Fine sediment, high organic content
[14]	India	Sediment	*tetM*, *bla_TEM*	10^2^–10^5^ copies/g	16S + qPCR	Human-impacted mangrove
[13]	China	Water & Sediment	*tetA*, *sul1*, *bla_CTX-M*, *mdt*	10^3^–10^6^ copies/g	Metagenomics	High MGE content
[4]	Global	Sediment	Various ARGs	10^2^–10^6^ copies/g	Metagenomics	Global sewage comparison

^1^ ARG abundance: number of copies per gram sediment or per mL water (as reported in original studies). ^2^ Notes describe environmental or methodological context: Aquaculture-influenced: Sites near aquaculture operations. Urban runoff: Areas affected by urban wastewater input. High MGE content: Samples with high prevalence of mobile genetic elements (plasmids, integrons, transposons). Human-impacted mangrove: Mangrove areas influenced by human activities. Global sewage comparison: Reference studies comparing ARGs to global sewage resistomes.

**Table 2 antibiotics-14-01022-t002:** Distribution and prevalence of antibiotic resistance genes (ARGs) in mangrove and coastal environments.

ARG Class	Representative Genes	Reported Occurrence ^1^ (%)	Notes/Sample Type	References
Tetracycline	*tetA*, *tetM*	40–55	Water, sediment, mangrove sediments	[1,11,14]
Sulfonamide	*sul1*, *sul2*	25–35	Urban watershed, community water	[4,6]
β-lactamase	*blaTEM*, *blaSHV*, *blaCTX-M*	20–30	Water, sediment, Enterobacteriaceae	[7,9]
Multidrug	*mdtK*, *acrB*	15–25	Mangrove sediments, aquaculture	[1,16]
Aminoglycoside	*aac(3)-II*, *aph(3′)-III*	10–20	Mangrove sediments	[3,11]
Macrolide	*ermB*, *mefA*	5–15	Sediment, human-associated samples	[10,15]
Quinolone	*qnrS*, *qnrB*	5–10	Coastal sediments	[2,5]

^1^ Reported prevalence: proportion of samples with detectable ARGs, as reported in original studies.

## Data Availability

All data used in this review are from previously published studies and are available in the cited references.

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
