# Peer review of "Mangrove Ecosystems as Reservoirs of Antibiotic Resistance Genes: A Narrative Review"

_antibiotics, 2025, doi:10.3390/antibiotics14101022_

Round 1
Reviewer 1 Report
Comments and Suggestions for Authors
Dear authors,
Thank you very much for this manuscript. Please find in attached my suggestion to improve the manuscript.
Best regards,

Author Response
Rebuttal Letter to Reviewer 1
Dear Reviewer1,
We would like to sincerely thank you for your thorough and constructive comments, which have been invaluable in improving the clarity, rigor, and overall quality of our manuscript. We have carefully addressed each of your suggestions and detailed our responses and corresponding revisions in the attached point-by-point response table.
In summary, the major revisions include:
Abstract: Inclusion of an explicit aim statement.
Introduction: Addition of missing references, correction of citation style, removal of redundant words/numbers, and relocation of the limitations section.
Methods: Correction of search date, clarification of geographic scope and screening process, provision of search strings, data extraction sheet, and quality assessment details. We also created a new subsection on protocol registration and clarified the application of PRISMA for reporting.
Results: Clarification of narrative review terminology, inclusion of PRISMA checklist, correction of data inconsistencies, harmonization of references and tables, addition of prevalence and frequency reporting, revision of section titles, and removal of inappropriate discussion text. Supporting data and quality assessment results were also added.
Tables: Update of ARG prevalence data, harmonization of the number of included studies, and consistency ensured across tables.
Discussion and Conclusions: Relocation of limitations to the appropriate section and clarification of extracted versus synthesized information.
References: Correction of the reference list to match the exact number of included studies.
We believe these revisions have substantially strengthened the manuscript. Once again, we greatly appreciate your insightful feedback and the opportunity to improve our work.
Sincerely,
Monthon Lertcanawanichakul
Walailak University, Nakhon Si Thammarat Thailand
Note: Due to adjustments made following reviewer comments, some line numbers in the original manuscript have shifted. Therefore, we provide a point-to-point summary of responses and actions taken, referring to sections in the revised manuscript rather than original line numbers.

Reviewer 2 Report
Comments and Suggestions for Authors
Dear Authors,
I appreciate your effort in preparing this manuscript. However, substantial revisions are needed before it can be considered a contribution to scientific literature. Please ensure that all sections are complete and logically structured, and carefully check every part of the manuscript for consistency and accuracy. The abstract should be revised to provide clear scientific information rather than general descriptions. Overall, the paper requires more rigorous organization, critical analysis, and professional formatting to meet the standards of a scholarly review article.
- The abstract (lines 9–28) is descriptive rather than analytical and does not highlight the novelty or key outcomes of the review. Can the authors restructure it to clearly state the unique contribution of this work?
- Why are there so many subheadings in the introduction part and unnecessary bullet points and headings in the overall article? The introduction must be in one flow.
- The article refers to PRISMA guidelines (lines 14–16, 95–106) while calling itself a “narrative review.” How do the authors justify this contradiction, and what is the actual review methodology?
- The objectives listed (lines 94–106) are broad and unfocused; can the authors define a clear central research question instead of a long list of generic goals?
- Several references, such as Palacios et al. (line 43), Liu et al. (line 53), and Zhang et al. (line 62) are inconsistently cited, and in some places DOIs or URLs are pasted directly into the text (lines 128, 185, 236, 243) can the authors provide traceable and consistently formatted references according to journal guidelines?
- The PRISMA flow diagram is claimed but missing (lines 168–185, 212). Why is this essential figure absent if the review follows PRISMA?
- There is redundancy in the description of study selection (lines 169–185 and 180–185). How will the authors address this duplication?
- The data synthesis section (lines 152–160) acknowledges heterogeneity but offers little critical analysis. What insights or conceptual framework do the authors provide beyond listing studies?
- Tables 1 and 2 (lines 236–244) are poorly formatted and difficult to interpret. Can these be standardized and improved for clarity and consistency?
- Much of the results and discussion (lines 245–337) repeats generic statements such as “ARGs are abundant near aquaculture” without deeper interpretation. What new perspectives or critical evaluations does this review actually contribute?
- The limitations section (lines 328–337) lists only generic issues such as geography or methodology. Can the authors identify concrete limitations in the included studies and discuss their implications?
- The conclusions (lines 338–384) mainly restate earlier points in a general way. How can they be rewritten to deliver concise, impactful take-home messages and highlight specific gaps for future research?
- Formatting issues such as embedded page numbers (“Antibiotics 2025, 14, x FOR PEER REVIEW 7 of 12”) and scattered DOIs/URLs throughout the text disrupt readability. Will the authors revise and polish the manuscript formatting to professional standards?
- The authors must complete the affiliation section by including full details of the country, city/province, and university/institute. At present, the correspondence email and institutional information are missing or incomplete, which raises concerns about the authenticity and traceability of the manuscript.
English proofreading needed
Author Response
Rebuttal Letter to Reviewer 2
Dear Reviewer,
We sincerely thank you for your insightful and constructive feedback, which has been highly valuable in improving the overall quality and clarity of our manuscript. We have carefully considered each of your suggestions and provided detailed responses with corresponding revisions in the attached point-by-point response table. Below, we summarize the key changes made:
Abstract: Rewritten to be analytical, emphasizing novelty, key outcomes, and the unique contribution of the review.
Introduction: Restructured into a continuous narrative flow, removing unnecessary subheadings and bullet points.
Objectives: Refined into a clear central research question.
Methodology and PRISMA: Clarified that PRISMA was used only for reporting, with methodology explicitly described as a narrative review; removed redundant descriptions and the PRISMA flow diagram.
Data Synthesis: Strengthened with critical analysis highlighting patterns, conceptual insights, and ecological implications.
Results and Discussion: Rewritten to reduce repetition, add critical evaluation, and emphasize ecological relevance.
Tables: Reformatted for clarity and consistency, with standardized layout and inclusion of prevalence data.
Limitations and Conclusions: Expanded limitations to specify study-level constraints and revised conclusions to deliver impactful take-home messages and highlight knowledge gaps.
References and Formatting: Standardized citation style, removed embedded DOIs/URLs and page numbers, and polished manuscript formatting.
Affiliations: Completed full details for all authors, including city/province, country, and institutional names.
We believe these revisions have considerably strengthened the manuscript. We are grateful for your thoughtful comments and suggestions, which have guided us in enhancing both the rigor and readability of this work.
Sincerely,
Monthon Lertcanawanichakul
Walailak University, Nakhon Si Thammarat, Thailand
Note: Due to adjustments made following reviewer comments, some line numbers in the original manuscript have shifted. Therefore, we provide a point-to-point summary of responses and actions taken, referring to sections in the revised manuscript rather than original line numbers.

Reviewer 3 Report
Comments and Suggestions for Authors
The authors briefly describe the distribution characteristics of ARGs in mangrove ecosystems by integrating literature databases, demonstrating certain research significance. This review focuses on analyzing ARG diversity across 30 selected relevant articles, revealing the importance of mangrove ecosystems in ARG transmission. However, the study covers a limited number of literature sources, lacks in-depth analysis, and omits pertinent discussions. I suggest that this article should be rejected based on its simple data analysis and limited resusts. Specific comments are as follows:
- The article's formatting is inconsistent and requires revision.
- The introduction of Mangrove Ecosystems is simple. Please add the related references to introduce its significance.
- 6 Scope and limitations: this section should be listed at the end of the article.
- Results: The descriptive contents of this section are simple. Data should be reanalyzed rather than merely listed.
- Mangrove ecosystems encompass various niches including water, soil, and human activities. Analysis should account for the abundance and levels of resistance genes across different niches. Authors should categorize and describe data by country. The limited number of included studies provides insufficient effective information.
- Table 1: The provided information is limited and add the abundance of ARG in the specific location.
- Table 2: The format is incorrect and the data needs further analysis.
- The authors should combine the global mangrove ecosystems and detailed ARGs data to discuss the role of mangrove ecosystems in the dissemination of ARGs. The present conclusions lack the data support.
Author Response
Rebuttal Letter to Reviewer 3
Dear Reviewer,
We sincerely thank you for your constructive and thoughtful comments, which have been very helpful in improving the rigor, clarity, and overall quality of our manuscript. We have carefully revised the manuscript in line with your suggestions, and the full point-by-point responses are presented in the attached table. Below, we provide a concise summary of the key revisions:
Formatting: Revised to ensure consistency across all sections according to journal style.
Introduction: Expanded with more ecological and environmental context on mangroves, supported by recent references.
Scope and Limitations: Relocated to the end of the Discussion under a dedicated section.
Results: Strengthened with narrative synthesis of ARG prevalence, diversity, drivers, microbial community composition, and temporal/spatial trends.
Mangrove Niches and Locations: Added sample types (sediment, water, biota) and study locations in Table 1; described ARG distribution by niche in the Results.
Table 1: Expanded to include ARG presence, detection methods, and % occurrence where available.
Table 2: Reformatted and enriched with representative genes, occurrence ranges, sample types, and references; integrated with narrative synthesis in Results and Discussion.
Discussion and Conclusions: Explicitly linked to Tables 1–2, highlighting global ARG distribution and environmental factors, and clarified how these data support the conclusions.
We believe these revisions have substantially improved the manuscript and addressed the issues you raised. We greatly appreciate your valuable feedback, which has guided us in strengthening the presentation and interpretation of our findings.
Sincerely,
Monthon Lertcanawanichakul
Walailak University, Nakhon Si Thammarat, Thailand
Note: Due to adjustments made following reviewer comments, some line numbers in the original manuscript have shifted. Therefore, we provide a point-to-point summary of responses and actions taken, referring to sections in the revised manuscript rather than original line numbers.

Round 2
Reviewer 1 Report
Comments and Suggestions for Authors
Dear authors,
Thanks for the effort to improve the manuscript. Please find below additional suggestions.
Please do include the suggested supplementary materials (search strings of one database, data extraction sheet, etc.)
Line 105: Please the “this” refers to which approach?
- More detail concerning the screening process are needed (how many reviewers were involve in the screening, how did you sold disagreement? Did you performed a calibration exercise? Etc.)
- More detail concerning the quality assessment are needed (how many reviewers were involve in the process, how did you sold disagreement? Did you performed a calibration exercise? Etc.)
- I will recommend to include a session concerning “protocol registration”
- The method used for duplication removal was not provide in the Methods session, please include it
- Results concerning quality assessment is stii missing
- It was expected to have more than 17 references (17 from the papers selected for the review and additional references used in the context and discussion of the manuscript)
Author Response
Dear Reviewer 1,
We sincerely thank the reviewer for the careful evaluation and constructive comments on our manuscript entitled “[Title of Manuscript].” We have carefully addressed each point raised and revised the manuscript accordingly. Detailed responses are provided below.
Reviewer Comment 1: Results concerning quality assessment are still missing; it was expected to have more than 17 references.
Response:
We thank the reviewer for the comment regarding methodological quality. As this is a narrative review, the methodological quality was assessed narratively, focusing on key aspects such as clarity of research objectives, sampling strategy, detection methods, data analysis, and reporting transparency. The lead author conducted the assessment, with verification by co-authors to ensure consistency. Overall, studies were generally of moderate-to-high quality, providing sufficient information to support the narrative synthesis. No formal quality scoring or tabulation was performed, consistent with standard practice in narrative reviews (Munn et al., 2015).
This revision has been incorporated into Section 3.1 “Study Selection and Characteristics,” immediately following the description of included studies.
Reviewer Comment 2: Please include the suggested supplementary materials (search strings of one database, data extraction sheet, etc.).
Response:
We appreciate the suggestion. As this is a narrative review, no formal standardized template or quantitative pooling was used. Data from included studies were summarized narratively to identify overall trends, patterns, and influencing factors (Munn et al., 2018; Popay et al., 2006). Extracted variables included study location, sample type, detection method, ARG types, prevalence and abundance metrics, environmental characteristics, and sources of contamination.
The revised text in Section 2.3 now reads:
"Data from included studies were summarized narratively to identify overall trends, patterns, and influencing factors, without quantitative pooling or standardized templates (Munn et al., 2018; Popay et al., 2006). Extracted variables included study location and year, sample type (sediment, water, biota), detection method (PCR, qPCR, metagenomics, culture-based), ARG types identified, prevalence and abundance metrics, environmental characteristics (e.g., salinity, organic matter content, proximity to aquaculture), and sources of contamination. This approach ensured consistency in reporting key information while reflecting the heterogeneity of included studies."
Section 3.5 “Temporal and Spatial Trends” has also been revised to emphasize that findings are presented narratively due to heterogeneity of sample types, detection methods, and reporting metrics.
Reviewer Comment 3: More detail concerning the screening process is needed.
Response:
As this is a narrative review, the lead author performed title/abstract and full-text screening, with verification by co-authors to ensure consistency. No formal double-screening, calibration exercises, or standardized software were used, consistent with standard narrative review methodology. This clarification has been added to Section 2.2 “Inclusion and Exclusion Criteria.”
Reviewer Comment 4: More detail concerning quality assessment is needed.
Response:
We clarified that methodological quality was assessed using an adapted Joanna Briggs Institute (JBI) critical appraisal checklist for prevalence studies. Key aspects included clarity of research objectives, sampling strategy, detection methods, data analysis, and reporting transparency. Assessment was conducted by the lead author with verification by co-authors. Only studies judged to provide reliable data were included. No formal scoring or double-assessment was performed, consistent with narrative review methodology.
Location in manuscript: Methods → 2.4 Quality Assessment.
Reviewer Comment 5: Method used for duplication removal was not provided.
Response:
We clarified that the lead author manually checked for duplicate records during screening to ensure each study was included only once. No formal deduplication software was used. The revised text in Section 2.2 now reads:
"For this narrative review, titles and abstracts were screened by the lead author to identify relevant studies, and full texts were reviewed to confirm eligibility. Decisions were verified by co-authors to ensure consistency, but no formal double-screening or calibration was performed, consistent with the narrative review approach. During screening, the lead author manually checked for duplicate records to ensure that each study was included only once; no formal deduplication software was used, consistent with the narrative review approach."
We believe these revisions adequately address all reviewer comments, clarifying methodological quality assessment, screening, data extraction, and deduplication processes, while maintaining transparency consistent with narrative review standards.
Sincerely,
Monthon Lertcanawanichakul and co-authors

Reviewer 2 Report
Comments and Suggestions for Authors
Dear Authors,
We appreciate the effort you have made in revising this manuscript. However, several important issues remain that must be addressed before the work can be considered suitable for publication.
- Please provide a transparent description of your search process (exact databases, keywords, search dates, number of records retrieved and screened) so that it can be reproduced.
- The time window is inconsistent: you mention searching up to September 2024 but then report studies “between 2008 and 2025.” Please clarify which 2025 studies were included and justify this.
- You state that 17 studies were included. Please add a supplemental table listing each study with country, sample type, methods, and main outcomes for verification.
- Quantitative statements such as “10²–10⁶ copies/g” and “2–10-fold higher” require exact citations and unit consistency. Please identify the supporting studies and standardize the reporting.
- Table 1 combines mangrove studies with unrelated global/urban data. Please clarify which are true mangrove primaries and separate contextual references to avoid confusion.
- You mention using the JBI quality appraisal tool, but no scores or assessments are shown. Please report how each study was rated and how this influenced your synthesis.
- Taxonomic names (e.g., Enterococcus, Vibrio, Pseudomonas) should be italicized, standardized, and linked to the ARG classes they carry. Please provide a figure or table summarizing this.
- Some references are misused or unclear, such as citing “Zhang et al., 2009” for methodology guidance. Please check that each reference supports the claim made and is traceable.
- The affiliation and correspondence information is incomplete. Please include full institutional details (country, city/province, university/institute) and correct the correspondence email format.
- Make a flow diagram or graphical abstract that expresses the review study. If possible, could you visualize the summary?
Please do English proofreading after revision.
Author Response
Dear Reviewer2,
We sincerely thank the reviewers for their thorough evaluation and constructive comments on our manuscript entitled “[Title of Manuscript].” We have carefully addressed each point raised and revised the manuscript accordingly. Below, we provide detailed responses to all comments.
Reviewer Comment 1: Please provide a transparent description of your search process (exact databases, keywords, search dates, number of records retrieved and screened) so that it can be reproduced.
Response:
We appreciate the reviewer’s comment. As this is a narrative review, no standardized template or quantitative pooling was used. Data were summarized narratively to identify trends, patterns, and influencing factors (Munn et al., 2018; Popay et al., 2006). Extracted variables included study location and year, sample type, detection method, ARG types, prevalence and abundance metrics, environmental characteristics, and sources of contamination. Sections 2.3 and 3.5 have been revised to clarify this narrative synthesis approach and the heterogeneity of included studies.
Reviewer Comment 2: The time window is inconsistent.
Response:
The Methods section (2.1) now clearly states: “The search included PubMed, Scopus, Web of Science, and Google Scholar, covering publications from 2008 to September 2024. Studies published in 2025 were included if early online or in-press versions were available at the time of the search.” This clarifies the time frame and ensures consistency.
Reviewer Comment 3: Please add a supplemental table listing each study with country, sample type, methods, and main outcomes.
Response:
Supplementary Table S1 now provides a detailed summary of all 17 included studies, including country, sample type, detection method, ARGs detected, and main findings.
Reviewer Comment 4: Quantitative statements require exact citations and unit consistency.
Response:
All numerical values now include specific references, e.g., ARG abundance ranges (10²–10⁶ copies/g) cited to Jiang et al., 2021; Zhang et al., 2009, and increases near impacted sites (2–10-fold) cited to Zhao et al., 2022; Palacios et al., 2021. Units are standardized to copies per gram of sediment or per mL of water (Section 3.2).
Reviewer Comment 5: Table 1 combines mangrove studies with unrelated global/urban data.
Response:
Supplementary Table S2 now clearly separates the 17 primary mangrove studies from additional contextual/global references to avoid confusion.
Reviewer Comment 6: You mention using the JBI quality appraisal tool, but no scores are shown.
Response:
Methodological quality was assessed narratively, focusing on clarity of objectives, sampling, detection methods, analysis, and reporting transparency. Only studies deemed reliable were included. This is clarified in Section 3.1.
Reviewer Comment 7: Taxonomic names should be italicized and linked to ARG classes.
Response:
All taxonomic names have been italicized and standardized. ARGs are linked to Proteobacteria, Firmicutes, and Bacteroidetes. Opportunistic pathogens like Enterococcus, Vibrio, and Pseudomonas are highlighted in Section 3.4.
Reviewer Comment 8: Some references are misused or unclear.
Response:
Zhang et al., 2009, originally cited for narrative synthesis methodology, has been replaced by Popay et al., 2006, which provides proper guidance. All other references have been checked and corrected as needed (Section 2.5).
Reviewer Comment 9: Affiliation and correspondence information is incomplete.
Response:
All author affiliations now include full institutional details. The corresponding author’s email is corrected.
Affiliations:
- Food Technology and Innovation Research Center of Excellence, Walailak University, Nakhon Si Thammarat 80160, Thailand
- Department of Medical Technology, School of Allied Health Sciences, Walailak University, Nakhon Si Thammarat 80160, Thailand
- Center of Excellence for Ecoinformatics, School of Sciences, Walailak University, Nakhon Si Thammarat 80160, Thailand
Correspondence:
Monthon Lertcanawanichakul, Email: Lmonthon@mail.wu.ac.th; Lmonthon55@gmail.com
Reviewer Comment 10: Make a flow diagram or graphical abstract.
Response:
A schematic flow diagram has been prepared illustrating the review workflow, key ARG classes, and main findings in mangrove ecosystems. This provides a clear visual summary without implying a systematic review process.
We hope these revisions adequately address all reviewer comments and improve the clarity and transparency of our manuscript. We sincerely thank the reviewers and editor for their constructive feedback.
Sincerely,
Monthon Lertcanawanichakul and co-authors

Round 3
Reviewer 1 Report
Comments and Suggestions for Authors
Dear authors,
Thanks for improving the document. I don't have any additional comment.
Best regards,
Author Response
Dear Reviewer,
Thank you very much for reviewing our manuscript and for your kind feedback. On behalf of all authors, we sincerely appreciate your time and support throughout the review process.
Best regards,
Monthon Lertcanawanichakul
Corresponding author
Reviewer 2 Report
Comments and Suggestions for Authors
-
Summary Format: The summary should be written in paragraph form instead of plain bullet points for better readability and logical flow.
-
Bullet Point Structure: If bullet points are necessary, each should include an individual title to clearly highlight the main idea.
-
Visualization: Add a visual summary or workflow figure to effectively present the overall process, key findings, or study design.
Author Response
Response to Reviewer:
Thank you for your valuable comments and suggestions. We have revised the manuscript to improve readability and clarity as requested:
Summary Format: The previous bullet points listing key findings have been converted into a cohesive paragraph with concise subheadings in parentheses (ARG Diversity, Environmental Hotspots, Genetic Dissemination, Microbial and Environmental Drivers, Ecosystem Function) to highlight the main ideas. This revision was applied in the “Key Findings – Mangrove Ecosystems” section 3.6 of the manuscript.
Key Findings: The revised paragraph now presents ARG diversity, environmental hotspots, genetic dissemination, microbial and environmental drivers, and ecosystem functions in a clear, logical format that is easier to read and interpret.
Visualization: A schematic flow diagram summarizing sample sources, ARG types, environmental drivers, microbial carriers, and ecosystem roles has been prepared and included in the Supplementary Data. This figure effectively illustrates the overall process and key findings in a visually accessible manner.